# VDR and PDIA3 Are Essential for Activation of Calcium Signaling and Membrane Response to 1,25(OH)_2_D_3_ in Squamous Cell Carcinoma Cells

**DOI:** 10.3390/cells13010011

**Published:** 2023-12-20

**Authors:** Joanna I. Nowak, Anna M. Olszewska, Justyna M. Wierzbicka, Magdalena Gebert, Rafał Bartoszewski, Michał A. Żmijewski

**Affiliations:** 1Department of Histology, Medical University of Gdansk, 80-211 Gdansk, Poland; j.chorzepa@gumed.edu.pl (J.I.N.); anna.olszewska@gumed.edu.pl (A.M.O.); justyna.wierzbicka@gumed.edu.pl (J.M.W.); 2Department of Medical Laboratory Diagnostics-Fahrenheit Biobank BBMRI.pl, Medical University of Gdansk, 80-134 Gdansk, Poland; magdalena.gebert@gumed.edu.pl; 3Department of Biophysics, Faculty of Biotechnology, University of Wroclaw, 50-383 Wroclaw, Poland; rafal.bartoszewski@uwr.edu.pl

**Keywords:** PDIA3, 1,25-dihydroxy vitamin D_3_ signaling, VDRE, NFAT, Ca^2+^ signaling

## Abstract

The genomic activity of 1,25(OH)_2_D_3_ is mediated by vitamin D receptor (VDR), whilst non-genomic is associated with protein disulfide isomerase family A member 3 (PDIA3). Interestingly, our recent studies documented that PDIA3 is also involved, directly or indirectly, in the modulation of genomic response to 1,25(OH)_2_D_3_. Moreover, PDIA3 was also shown to regulate cellular bioenergetics, possibly through the modulation of STAT signaling. Here, the role of VDR and PDIA3 proteins in membrane response to 1,25(OH)_2_D_3_ and calcium signaling was investigated in squamous cell carcinoma A431 cell line with or without the deletion of *VDR* and *PDIA3* genes. Calcium influx was assayed by Fura-2AM or Fluo-4AM, while calcium-regulated element (NFAT) activation was measured using a dual luciferase assay. Further, the levels of proteins involved in membrane response to 1,25(OH)_2_D_3_ in A431 cell lines were analyzed via Western blot analysis. The deletion of either *PDIA3* or *VDR* resulted in the decreased baseline levels of Ca^2+^ and its responsiveness to 1,25(OH)_2_D_3_; however, the effect was more pronounced in A431∆*PDIA3*. Furthermore, the knockout of either of these genes disrupted 1,25(OH)_2_D_3_-elicited membrane signaling. The data presented here indicated that the VDR is essential for the activation of calcium/calmodulin-dependent protein kinase II alpha (CAMK2A), while PDIA3 is required for 1,25(OH)_2_D_3_-induced calcium mobilization in A431 cells. Taken together, those results suggest that both VDR and PDIA3 are essential for non-genomic response to this powerful secosteroid.

## 1. Introduction

The classical signaling of 1,25(OH)_2_D_3_ is mediated by the nuclear vitamin D receptor (VDR). This nuclear receptor for vitamin D, together with its co-receptor protein, retinoid X receptor (RXR), forms heterodimers, forming a powerful transcription factor. The VDR-RXR dimer upon ligand binding is translocated into the nucleus, where it binds to vitamin D response elements (VDRE) in the regulatory region of the vitamin D target genes and modulates their expression [1,2]. One of the principal activities of 1,25(OH)_2_D_3_ is the regulation of calcium-phosphate homeostasis [3,4].

However, not all the effects of 1,25(OH)_2_D_3_ can be attributed to genomic response; thus, the idea of alternative signaling pathways with potential membrane-bound receptor for this secosteroid has emerged [5]. The presence of such a pathway, for example, explained the rapid influx of calcium ions induced with 1,25(OH)_2_D_3_ [6,7]. Eventually, Nemere and coworkers described a 1,25D_3_-membrane-associated, rapid response steroid-binding protein (1,25D_3_-MARRS), which was also known as protein disulfide isomerase family A member 3 (PDIA3) [8]. PDIA3 is an endoplasmic reticulum (ER) protein involved in protein folding, together with other chaperones like calnexin or calreticulin [9]. Outside of ER, PDIA3 was localized within the cell membrane, nucleus, cytoplasm, or mitochondria [10]. Importantly, this protein was proven to be involved in a rapid uptake of calcium and phosphate in intestine cells induced by 1,25(OH)_2_D_3_ [11,12]. Thus, PDIA3 is strongly linked to calcium homeostasis. Lately, it has been shown that PDIA3 knockout in squamous cell carcinoma alters the expression of the genes connected to the regulation of bone mineralization, phospholipase C activity, and calcium-dependent phospholipid binding [13]. Moreover, it was shown that the partial silencing of *PDIA3* (*PDIA3*^+/−^) in mice model impaired skeletal development, while deletion was lethal [14,15,16].

PDIA3 was shown to interact with the phospholipase A2-activating protein (PLAA), subsequently leading to the activation of phospholipase A2 (PLA2) and the mediation of the non-genomic rapid response to 1,25(OH)_2_D_3_. As a result, calcium is released to the cytoplasm, followed by the activation of protein kinase C (PKC) or calcium/calmodulin-dependent protein kinase II (CaMKII). Thus, this leads to the induction of downstream signaling such as mitogen-activated protein kinases (MAPK) pathways and other transcription factors (STAT1-3, NF-kB). It has been shown that the disruption of PDIA3 protein attenuated PKA, PKC signaling, and calcium influx [11,14,17,18,19]. Recently, it has been shown that PDIA3 influences STAT3 expression in the *C. elegans* model, regulating cellular respiration [20]. Additionally, it has been shown that PDIA3 modulates STAT3 signaling induced by 1,25(OH)_2_D_3_ [21].

In our latest studies, we investigated the impact of deletion of either *VDR* or *PDIA3* on gene expression profile and non-genomic effects of 1,25(OH)_2_D_3_ in A431 squamous cell carcinoma, showing that PDIA3 is involved in genomic responses to 1,25(OH)_2_D_3_ [13,22]. Moreover, we have shown that the deletion of *PDIA3* abrogates the effects of 1,25(OH)_2_D_3_ on cellular bioenergetics, possibly through STAT3 signaling [21]. In this study, the effects of knockout of *PDIA3* and *VDR* on the 1,25(OH)_2_D_3_ membrane’s non-genomic signaling were investigated.

## 2. Materials and Methods

### 2.1. 1,25(OH)_2_D_3_

The active form of vitamin D (1,25(OH)_2_D_3_) was purchased from Sigma-Aldrich (Merck KGaA, Darmstadt, Germany). 1,25(OH)_2_D_3_ was dissolved in ethanol and stored at −20 °C. A 100 nM concentration was used in all experiments, as it was proven to have the most potent activity for proliferation as well as for photoprotection in other research [13,23,24].

### 2.2. Cell Cultures

Immortalized human basal cell carcinoma cell line (A431) was purchased from Synthego Corporation (Menlo Park, CA, USA). *PDIA3* and *VDR* knock-out cell lines were obtained with CRISPR/Cas9 technology, as described previously [13]. For cell cultures, DMEM high glucose medium (4.5 g/L) was used. Additionally, the medium was supplemented with 10% fetal bovine serum (FBS), penicillin (10,000 units/mL), and streptomycin (10 mg/mL) (Sigma-Aldrich; Merck KGaA, Darmstadt, Germany). A431 cell lines were cultured in the incubator with 5% CO_2_ at 37 °C. For experimental conditions, the medium was changed to DMEM supplemented with 2% charcoal-stripped FBS.

### 2.3. Measurement of Intracellular Calcium Concentration

A431 cell lines were seeded onto 96-well plates in DMEM medium supplemented with charcoal-stripped FBS and antibiotics. To test the effects of *VDR* and *PDIA3* deletion on calcium influx, cells were incubated with 1µM Fura-2AM solution (Sigma-Aldrich; Merck KGaA, Darmstadt, Germany) in HBSS for 1 h. Afterwards, incubation cells were rinsed with medium. The fluorescence intensity of the cells was measured using a plate reader at λ_ex_ 355 nm and λ_em_ 495 nm for 16 min.

A431 cell lines were seeded onto an 8-well chamber slide in DMEM medium supplemented with charcoal-stripped FBS and antibiotics. To test the effects of *VDR* and *PDIA3* deletion on calcium influx, cells were incubated with 1 µM Fluo-4AM solution (Sigma-Aldrich; Merck KGaA, Darmstadt, Germany) in Hank’s Balanced Salt solution (HBSS) (PAN Biotech, Aidenbach, Germany) for 30 min. Cells were rinsed with HBSS and left in medium. The intensity of fluorescence was observed with the use of live microscopy on Olympus Cell-Vivo IX 83 (Olympus, Tokyo, Japan).

### 2.4. Luciferase Reporter Assay

The human VDRE and NFAT firefly luciferase reporter constructs were purchased from Qiagen (Hilden, Germany). To test the effects of the deletion of either *VDR* or *PDIA3* on VDRE and NFAT elements, A431 cells were seeded onto 96-well polystyrene, white/clear flat bottom plates (30,000 cells/per well) and transfected with the constructs using Lipofectamin 2000 (Thermo Fisher Scientific, Waltham, MA, USA). After 24 h transfected cells were treated with 100 nM 1,25(OH)_2_D_3_ for 4, 8, or 24 h. Then, cells were lysed using luciferase assay lysis buffer (Promega, Madison, WI, USA), and the firefly-Renilla luciferase activities were measured using the Dual-Luciferase Reporter Assay (Promega, Madison, WI, USA) according to the manufacturer’s protocol using GloMax-Multi + Detection System (Promega, Madison, WI, USA). Results were normalized to non-treated cells for all A431 sublines, separately.

### 2.5. Western Blot

Squamous cell carcinoma cell lines were treated with 100 nM 1,25(OH)_2_D_3_ for 4, 8, and 24 h. After a given time (4 h, 8 h, and 24 h), the medium was removed and cells were washed twice with PBS. Next, A431 cells were scratched from the plate in cooled PBS, and the suspension was moved to an Eppendorf tube and centrifuged for 10 min at 16,000× *g*. The cell pellet was resuspended in RIPA buffer (Thermo Fisher Scientific, Waltham, MA, USA) with the addition of phosphatase and protease inhibitors (Sigma-Aldrich; Merck KGaA, Darmstadt, Germany). To measure the concentration of lysates, modified Bradford Assay was used according to manufacturer protocol (Bio-Rad, Hercules, CA, USA). Amounts of 10% bottom gel and 5% upper gel were used for SDS-PAGE electrophoresis. An equal amount of lysates (20 µg) was loaded into each well, and gels were resolved at 90–110 V in the Bio-Rad apparatus (Bio-Rad, Hercules, CA, USA). A Trans-Blot Turbo system was used for protein transfer to PVDF membranes (Bio-Rad, Hercules, CA, USA). Then, membranes were blocked in 5% milk dissolved in TBS-T. The membranes were incubated with specific primary antibodies: anti-PLAA, anti-PLCγ, anti-PKCα, anti-ERK1/2, anti-phophoERK1/2 (Thr202/Tyr204), anti-Caveolin 1, anti-Caveolin 3, anti-CAMK2A, anti-phosphoCAMK2A (T286), and anti-TRPV6 (Abclonal, Woburn, MA, USA), overnight at 4 °C. Proper secondary fluorescent antibodies conjugated with AlexaFluor^®^ 790 or AlexaFluor^®^ 680 were used (Jackson ImmunoResearch, Cambridgeshire, UK). As a loading control, anti-β-actin antibodies (Abclonal, Woburn, MA, USA) were used. Results were visualized with the Odyssey Clx system and calculated with the use of Image Studio Software Ver 5.2 (both LI-COR Biosciences, Lincoln, NE, USA).

### 2.6. Bioinformatic Analysis

As described in a previous study [13,22], data quality and cell line disparity were checked. To identify differentially expressed genes, the absolute value of log2fold change ≥ 1.0 and adjusted *p*-value < 0.05 were used. To see whether the expression of selected genes was regulated by either the deletion of *PDIA3* or 1,25(OH)_2_D_3_ treatment, the heat maps were prepared. Data showed the clustering of RNA-seq expression data and technical repeats within the groups. The RNA-seq data have been deposited in Sequence Read Archive (SRA) under accession number PRJNA926032.

### 2.7. Statistical Analysis

GraphPad Prism version 7.05 was used for the statistical analysis of obtained data (GraphPad Software, Inc., La Jolla, CA, USA). Results are presented as mean ± SD and were analyzed either with Student’s *t*-test or one-way ANOVA analysis of variance with appropriate post hoc tests. Statistical significance is illustrated as asterisks: * *p* < 0.05, ** *p* < 0.01, *** *p* < 0.001, or **** *p* < 0.0001.

## 3. Results

### 3.1. Deletion of PDIA3 Modulates the Expression of Calcium-Associated Genes in A431 Cells

Previously, it was shown that deletion of the *PDIA3* gene in the A431 squamous cell carcinoma line (A431*ΔPDIA3*) affected expression of more than 1800 genes, including genes modulated by 1,25(OH)_2_D_3_ treatment [13]. Here, the expression of calcium-associated genes and ER-related genes was evaluated. Thirty calcium-associated genes were found amongst differently expressed genes (DEGs) in the A431*ΔPDIA3* cell line, including 12 upregulated and 18 downregulated genes (Figure 1A). All of those DEGs associated with calcium were further affected by 1,25(OH)_2_D_3_ treatment, and 14 were upregulated and 16 downregulated in the *PDIA3* knockout cell line (Figure 1B). Further, those 30 calcium-associated DEGs disrupted by *PDIA3* deletion were analyzed in terms of the biological process using gene ontology. The analysis revealed their involvement in biological processes related to the positive regulation of mitochondrial calcium ion concentration, mitochondrial calcium ion homeostasis, the negative regulation of endoplasmic reticulum calcium ion concentration, and the positive regulation of calcium ion transport (Figure 1C).

### 3.2. Deletion of PDIA3 Decreases Calcium Levels after 1,25(OH)_2_D_3_ Treatment and Consequently Disrupts the Activity of Calcium-Regulated Nuclear Factor of Activated T-Cells (NFAT) in A431 Cells

Subsequent experiments were focused on changes in levels of intracellular calcium induced by 1,25(OH)_2_D_3_. To verify an impact of PDIA3 on calcium homeostasis, intracellular levels of calcium were measured with the use of two fluorescence probes (Fura-2AM and Fluo-4AM) via fluorescent measurement using plate reader or live microscopy on Olympus Cell-Vivo IX 83, respectively (Figure 2). In A431 knockout cell lines, the baseline level of intracellular calcium measured with Fluo-4AM probe was decreased in comparison to A431WT cells. The effect was less pronounced for A431∆*VDR* cells. Curiously, in the case of A431∆*PDIA3* cells, there was a decrease in fluorescence after 1,25(OH)_2_D_3_ addition (Figure 2A,B). Those results are further supported by data acquired with a Fura-2AM probe, indicating a role of PDIA3 in calcium mobilization and 1,25(OH)_2_D_3_-induced calcium influx. An addition of 100 nM 1,25(OH)_2_D_3_ elicited an influx of calcium ions into the A431 cells; however, the increase in both cell lines, A431∆*VDR* and A431∆*PDIA3*, was significantly smaller in comparison to A431WT cells. A431∆*PDIA3* cells were characterized by the lowest baseline calcium level and 1,25(OH)_2_D_3_-induced calcium influx. Additionally, lower concentrations of 1,25(OH)_2_D_3_ (1 nM and 10 nM) were tested with a similar effect (Appendix A), suggesting that even low concentrations of 1,25(OH)_2_D_3_ are sufficient for the activation of calcium influx in A431 cells. This is in agreement with previous studies where calcium uptake was triggered by even lower, picomolar concentrations of 1,25(OH)_2_D_3_ (0.13 nM [7] or 0.3 nM [25]). Calimycin (calcium ionophore) was used as a positive control of calcium influx. Interestingly, a calimycin-induced influx of calcium ions was also slightly affected by either *VDR* or *PDIA3* deletion in A431 cells and the effect was more pronounced in Δ*PDIA3* cells (Figure 2C,D). In our recent publication, it was shown that *PDIA3* deletion impacts the expression of the well-known calcium channel TRPV6 [13]. Here, the level of the mentioned protein was checked after the deletion of *VDR* or *PDIA3* and 1,25(OH)_2_D_3_ treatment. In A431WT cells 1,25(OH)_2_D_3_ treatment increased levels of TRPV6 protein. The same effect was observed for A431∆*VDR* cells, but the increase was slightly reduced in comparison to the A431WT cell line. Interestingly, the deletion of *PDIA3* prevented the increase in TRPV6 protein (Figure 2E). 

Since the transcriptome analysis revealed numerous changes in gene expression after *PDIA3* deletion [13] and, most importantly, among calcium-associated genes, the effects of *VDR* and *PDIA3* deletion on the activity of the calcium-associated nuclear factor of activated T-cells (NFAT) and vitamin D response elements (VDRE) were investigated. VDRE served as additional control. Briefly, after the transfection of A431WT, A431*∆VDR*, and A431*∆PDIA3* with NFAT or VDRE luciferase reporter vectors, cells were treated with 100 nM 1,25(OH)_2_D_3_ and luciferase activity was measured (Figure 3). The NFAT is a transcription factor activated by calcium signaling and was previously linked to the immunosuppressive activity of 1,25(OH)_2_D_3_ [26,27]. The NFAT activity was increased almost threefold in A431WT cells after 8 h of 1,25(OH)_2_D_3_ treatment (Figure 3A). The deletion of *VDR* did not affect NFAT activity (Figure 3B). Interestingly, in A431∆*PDIA3* cells, 1,25(OH)_2_D_3_ treatment elicited a rapid increase in NFAT activity after 4 h with further decrease in activity after 8 and 24 h (Figure 3C). In wild-type A431 cell line, 1,25(OH)_2_D_3_ treatment increased the activity of VDRE most efficiently after 24 h (Figure 3D). The deletion of *VDR* completely eliminated effect of 1,25(OH)_2_D_3_ treatment on vitamin D response element (Figure 3E). In contrast, *PDIA3* deletion slightly enhanced VDRE activation by 1,25(OH)_2_D_3_ in comparison to A431WT cells, and the effect was noticeable after 4 or 8 h of incubation (Figure 3F).

### 3.3. VDR and PDIA3 Deletion Disrupts Membrane Response to 1,25(OH)_2_D_3_ in Squamous Cell Carcinoma

Calcium is a known secondary messenger, which activates several downstream targets such as PKC or CAMKII. Thus, we decided to investigate the downstream targets of calcium signaling as a part of membrane response to 1,25(OH)_2_D_3_ and the role of VDR and PDIA3 proteins in this process. Thus, time-resolved analysis of the levels of calcium-related protein in A431 cell with deletion of either *VDR* or *PDIA3* was performed (Figure 4A). It was observed that *VDR* deletion alone increased baseline levels of the phosphorylated form of extracellular signal-regulated kinase 1/2 (pERK1/2; Thr202/Tyr204) while decreasing the phosphorylated form of calcium/calmodulin-dependent protein kinase II alpha (pCAMKIIα; T268) (Figure 4B,C). Interestingly, the deletion of *PDIA3* increased baseline levels Erk1/2. The total amount of CAMKIIα was not affected by either *VDR* or *PDIA3* deletion.

*Doroudi M* and coworkers have shown that PDIA3 is essential for the activation of PLAA, PLCγ, and PKCα signaling cascade by 1,25(OH)_2_D_3_ [18]. Although we observed that the baseline total levels of PLAA, PLCγ, and PKCα were increased by PDIA3 deletion (Appendix A), the change in PLA2 and PKCα activity measured by ELISA assays after 1,25(OH)_2_D_3_ stimulation was not observed on our cellular model (not shown). Finally, the knockout of the *PDIA3* gene disrupted the response of Erk1/2 and its phosphorylation after 1,25(OH)_2_D_3_ treatment (Figure 4C). *VDR* knockout also abolished the response of CAMKIIα and its phosphorylation (Figure 4B). Interestingly, 1,25(OH)_2_D_3_ treatment induced a slight increase in PDIA3 level after 24 h incubation, and *VDR* deletion enhanced this effect.

## 4. Discussion

Complementary to our previous research on the topic of non-genomic responses to 1,25(OH)_2_D_3_ in squamous cell carcinoma [13,21], we further explored the role of PDIA3 in calcium homeostasis and membrane signaling after 1,25(OH)_2_D_3_ treatment.

Recently, it has been shown that PDIA3 is strongly involved in the regulation of mitochondrial bioenergetics [20,21]. Here, we have shown that the deletion of the *PDIA3* gene affected the expression of calcium-associated genes and modulated their responses to 1,25(OH)_2_D_3_. Those genes were mainly involved in calcium-related processes within mitochondria and endoplasmic reticulum. Interestingly, the expression of some of those genes (*TGM2*, *BNIP3*, *FIS1*) has been linked as prognostic markers in squamous cell carcinomas [28,29]. The silencing of *PDIA3* and 1,25(OH)_2_D_3_ treatment reversed the expression of those cancer-related genes in A431 cells. In accordance with our previous study, we observed that deletion of *VDR* completely eliminates 1,25(OH)_2_D_3_-dependent gene expression [22]. Furthermore, we observed that in cells lacking VDR, the activation of the vitamin D response element was completely eliminated, while the deletion of *PDIA3* modulated the activity of VDRE to some extent, especially after 8 h of 1,25(OH)_2_D_3_ treatment.

Calcium acts as a second messenger molecule and is critical for proper cell physiology and signal transduction [30,31]. Here, we showed the deletion of either *VDR* or *PDIA3* decreased baseline calcium levels in squamous cell carcinoma A431 cells and further impaired calcium influx induced by 1,25(OH)_2_D_3_. Those findings were in accordance with our other study on PDIA3’s role in calcium signaling. He and coworkers showed that the knockdown of *PDIA3* inhibited mitochondrial calcium uptake within HeLa cells, possibly through the regulation of mitochondrial calcium uniporter (MCU) expression [32]. Additionally, it was shown that the reduction in VDR levels in the intestine blunts the 1,25(OH)_2_D_3_-regulated absorption of calcium [33]. Furthermore, it was shown in mice models that the transient receptor potential vanilloid type 6 (TRPV6) is essential for vitamin D-induced active calcium transport in the intestine [34]. TRPV6 is a well-known classical target for 1,25(OH)_2_D_3_ action and plays a vital role in the transcellular transport of calcium ions and uptake [35]. TRPV6 occurs in two forms, glycosylated (gTRPV6, 100 kDa) and non-glycosylated (TRPV6, 80 kDa). It was suggested that glycosylation determines the stability and assembly of TRPV6 [36]. In our previous research, it was shown that *PDIA3* deletion significantly impaired the expression of the *TRPV6* gene and its responsiveness to 1,25(OH)_2_D_3_ in A431∆*PDIA3* cells [13]. Bianco et al. have shown that the deletion of the TRPV6 calcium channel resulted in no response to PTH or 1,25(OH)_2_D_3_ treatment on mice models [37]. Here, we have shown that the expression of both forms of *TRVP6* was also impaired by *PDIA3* deletion in A431 cells. Decreased levels of TRPV6 in A431∆*PDIA3* may explain the partial impairment of calcium influx observed with a Fura-2AM probe. Thus, our results indicate that PDIA3 plays a major role in the regulation of calcium homeostasis, including the vitamin D-induced uptake of Ca^2+^ or intracellular storage and trafficking with the possible involvement of epithelial channel TRPV6 and the regulation of its stability.

PDIA3 is a protein necessary to maintain normal cellular physiology, and its dysfunction may lead to numerous diseases, including cancers, neurodegenerative diseases, or respiratory pathologies [38,39,40,41,42]. Multiple studies linked PDIA3 to the rapid response of cells to 1,25(OH)_2_D_3_ [17,18] but the exact mechanism of its action remains unclear. In our previous report, we showed that the deletion of the *PDIA3* gene strongly modulates the effect of 1,25(OH)_2_D_3_ on the gene expression profile of A431 cells, suggesting that it can directly (as a transcription factor or modulator) or indirectly (through activation of other signaling pathways and/or transcription factors [20,43,44]) affect the genomic activity of vitamin D. Among those genes, PDIA3 deletion increased the expression of PKCα [45]. Here, the total amount of PKCα was also disrupted and the effect of 1,25(OH)_2_D_3_ was reduced. This is in accordance with the study of Wang et al., where the deletion of PDIA3 impaired the 1,25(OH)_2_D_3_-induced activity of PKC [14]. Our results presented here demonstrate that both VDR and PDIA3 proteins are indeed involved in the 1,25(OH)_2_D_3_ membrane response of A431 cells [46]. Nevertheless, it seems that the cooperation of both is essential for membrane response. PDIA3 was co-localized with VDR in caveolae, and it was proven that both proteins can interact with caveolin-1 [17]. Another study by Doroudi and coworkers showed that CAMKIIA is required for mediating the rapid actions of 1,25(OH)_2_D_3_ [47]. However, in our study, only the deletion of *VDR* attenuated the impact of 1,25(OH)_2_D_3_ on levels of CAMKIIA. It was shown recently that the activation of PLCγ-mediated signaling results in the release of calcium from intracellular stores and induces an influx of ions across the plasma membrane [48]. Here, we observed that the deletion of *PDIA3* had the most prominent effect on the expression of PLCγ; however, a change in its activity was not observed, and thus, the involvement of PLCγ in 1,25(OH)_2_D_3_ signaling in the A431 squamous cell carcinoma cell line requires further investigation. Decreased levels of PKCα to 1,25(OH)_2_D_3_ in A431∆*PDIA3* cells are in accordance with the observations of Boyan and colleagues, who showed that the partial deletion of PDIA3 impaired the activation of PKC through PLAA induction due to lack of interaction between PDIA3/Cav1/PLAA [49].

Here, we showed a major role of PDIA3 in membrane response to an active form of vitamin D. Our results indicate that PDIA3 is not solely responsible for the activation of the non-genomic pathway of 1,25(OH)_2_D_3_, but that VDR is also required for this action. However, it seems that VDR and PDIA3 affect different targets of the 1,25(OH)_2_D_3_ membrane response. Our results are further supported by a previous study in which we stated that PDIA3 is a modulator of the genomic actions of vitamin D [13]. Moreover, it seems that VDR and PDIA3 are required for the regulation of calcium influx induced by 1,25(OH)_2_D_3_ in squamous cell carcinoma. The proposed involvement of PDIA3 and VDR in 1,25(OH)_2_D_3_ action is shown in Figure 5.

Taken together, our results presented here emphasize the importance of PDIA3 in 1,25(OH)_2_D_3_ signaling in squamous cell carcinoma cell line A431. The deletion of *PDIA3* not only affected membrane signaling but also genomic responses to vitamin D. Moreover, it seems that both VDR and PDIA3 are required for the regulation of calcium signaling induced by 1,25(OH)_2_D_3_ in A431 squamous cell carcinoma.

## 5. Conclusions

In conclusion, this study demonstrated that both VDR and PDIA3 are needed to regulate membrane response to active forms of vitamin D in A431 squamous cell carcinoma, possibly through CAMKIIα and impaired calcium influx, respectively. The proposed roles of PDIA3 and VDR in the regulation of intracellular response to 1,25(OH)_2_D_3_ are summarized in Figure 5.

## Figures and Tables

**Figure 1 cells-13-00011-f001:**
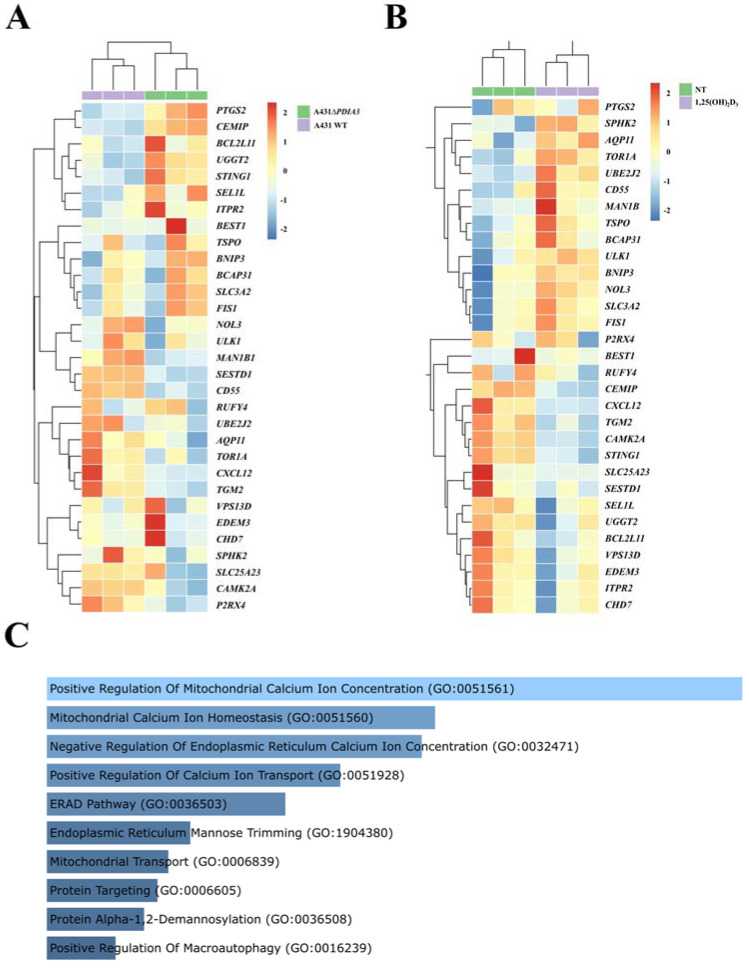
Changes in gene expression connected to calcium metabolism and endoplasmic reticulum homeostasis in human squamous carcinoma cell lines (A431). Heatmaps of selected genes with a statistically significant change in expression level after (**A**) PDIA3 knockout and (**B**) after 1,25(OH)_2_D_3_ treatment solely in A431∆*PDIA3* cells. The color of cells represents the Z-score of normalized gene expression values. (**C**) Gene ontology biological process analysis of calcium-associated genes disrupted after *PDIA3* deletion. Color intensity and length correspond to the *p*-value of the GO biological process.

**Figure 2 cells-13-00011-f002:**
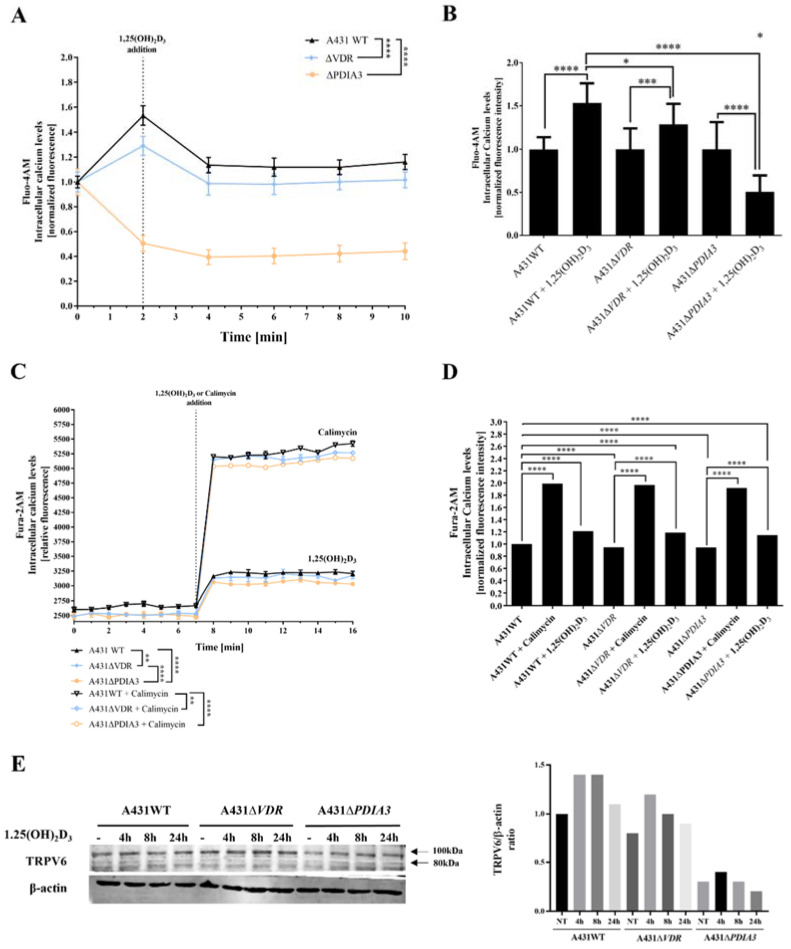
Deletion of *PDIA3* decreases calcium levels after 1,25(OH)_2_D_3_ treatment in squamous cell carcinoma. (**A**) Intracellular calcium levels monitored with Fluo-4AM fluorescence probe in A431 cell lines with live cell time-lapse microscopy. (**B**) Normalized fluorescence intensity at 0 and 2 min time points. (**C**) Time-resolved analysis of intracellular calcium levels measured with Fura-2AM probe on microplate reader in A431. Results were calculated as a mean ± SD of triplicates. Statistically significant differences are illustrated with asterisks: * *p* < 0.05, ** *p* < 0.01, *** *p* < 0.001, or **** *p* < 0.0001. (**D**) Alterations of normalized fluorescence intensity at the 7th and 8th minute of the experiment. Representative results were presented in the graph. (**E**) Analysis of TRPV6 protein levels in A431 cell lines with *VDR* or *PDIA3* deletion. Two bands represent glycosylated (100 kDa) and non-glycosylated (80 kDa). The quantity of non-glycosylated TRPV6 (80 kDa) was calculated as a protein/β-actin ratio. Protein levels are calculated as means from three independent experiments. Representative pictures are shown.

**Figure 3 cells-13-00011-f003:**
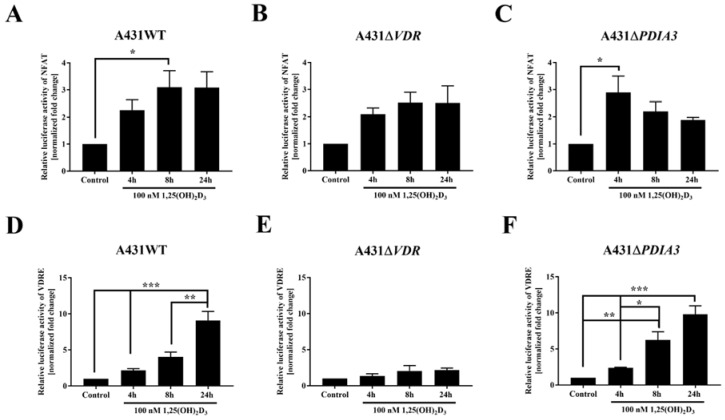
Changes in the activity of VDRE and NFAT response element to 1,25(OH)_2_D_3_ in A431 cell lines. The bar graphs indicate the normalized fold change of the firefly/*Renilla* luciferase ratio of NFAT (**A**–**C**) and VDRE (**D**–**F**) in A431WT, ∆VDR, and ∆PDIA3. Dual luciferase reporter assay results are represented as means ± SEM of triplicate samples. Statistically significant differences are illustrated with asterisks: * *p* < 0.05, ** *p* < 0.01, or *** *p* < 0.001.

**Figure 4 cells-13-00011-f004:**
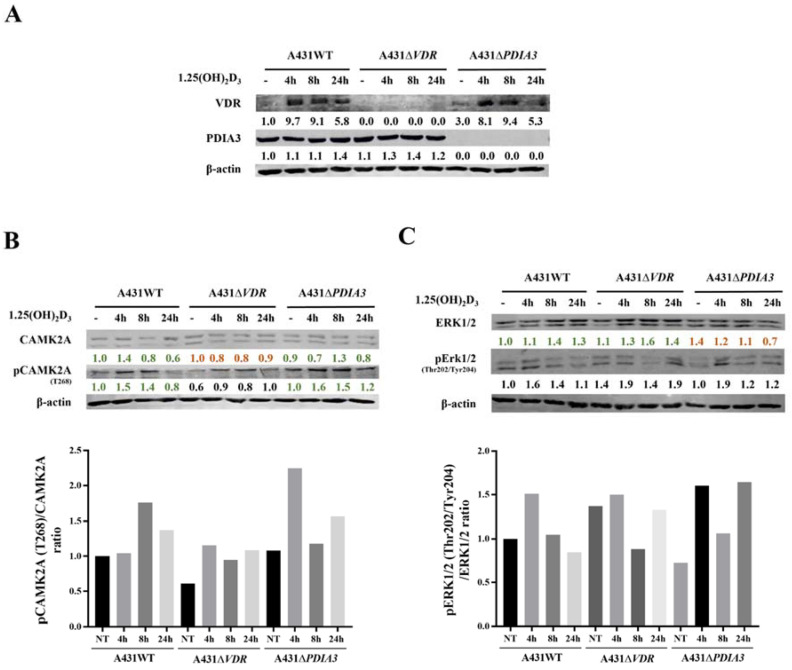
Effects of *PDIA3* and *VDR* deletion on 1,25(OH)_2_D_3_ signaling pathways in human squamous carcinoma cells (A431). (**A**) Validation of VDR and PDIA3 knockout in A431 cells. Ratio of phosphorylated forms of (**B**) CAMK2A (T286) and (**C**) Erk1/2 (Thr202/Tyr204) proteins. The red color illustrates a decrease in protein level, while green marks an increase. The quantity of each protein was calculated as a protein/β-actin ratio. Protein levels are calculated as means from three independent experiments. Representative pictures are shown for each protein. The red color illustrates a decrease in protein level, while green marks the increase.

**Figure 5 cells-13-00011-f005:**
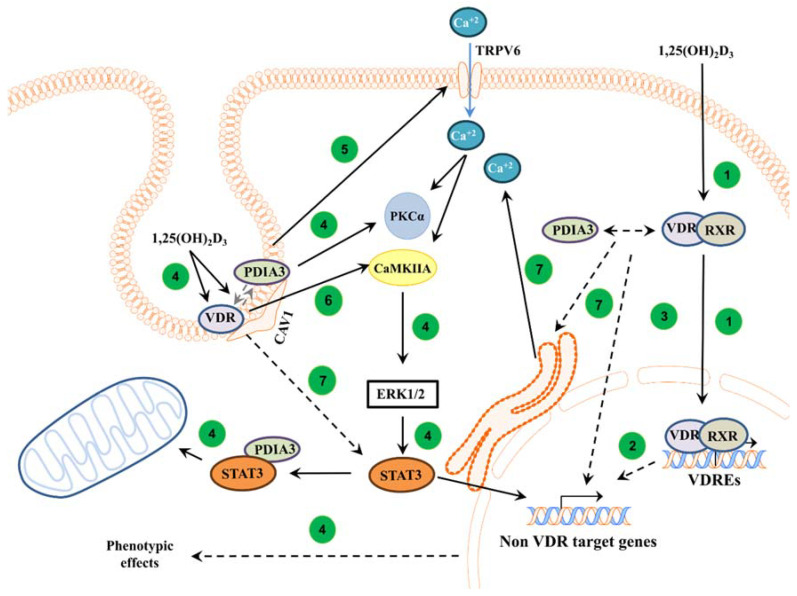
Proposed mechanism of action of VDR and PDIA3 to 1,25(OH)_2_D_3_ membrane response in squamous cell carcinoma. In the classical pathway, 1,25(OH)_2_D_3_ is bound by a heterodimer of VDR/RXR proteins and, subsequently, the complex is translocated into the nucleus where it regulates the transcription of vitamin D target genes [1] (1). Further, primary VDR target transcription factors can regulate secondary non-vitamin D target genes [50] (2). It was also postulated that PDIA3 can modulate genomic response to 1,25(OH)_2_D_3_ [13] (3). In non-genomic pathways, VDR and PDIA3 were shown to interact with caveolin-1 [17]. PDIA3 was shown to be essential to activate PKC after 1,25(OH)_2_D_3_ treatment [14] (4). Moreover, PDIA3 affects TRPV6 levels within SCC cells, possibly disrupting calcium response (5). Either PDIA3 or VDR are needed to activate STAT3, possibly regulating mitochondrial bioenergetics and non-VDR target genes [13,21] (4). Interestingly, it seems that VDR is required to activate CAMK2IIA kinase (6), while either protein is essential for valid calcium signaling (7).

## Data Availability

Data are contained within the article and Appendix A.

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
