# Peer review of "VDR and PDIA3 Are Essential for Activation of Calcium Signaling and Membrane Response to 1,25(OH)2D3 in Squamous Cell Carcinoma Cells"

_cells, 2023, doi:10.3390/cells13010011_

Round 1

Reviewer 1 Report

Comments and Suggestions for Authors

The paper by Nowak et al. describes the results of VDR or PDIA3 gene knock-outs in human basal cell carcinoma cell line (A431). The authors show that in comparison to wt-A431 cells, A431ΔPDIA3 cells respond differently to 1,25(OH)2D3 (the active form of vitamin D3). In this paper the authors concentrate on calcium transport and signalling.

Despite there are some interesting results in this paper, there are several issues that must be corrected before the paper is suitable for publication.

Major problems:

1.       1,25(OH)2D3 was used only at one, very high concentration of 100nM. Why so high concentration? Were lower concentrations tested as well?

2.       PDIA3 knock-out cell line was described in the Materials and methods, but VDR knock-out cell line was not.

3.       The description to Figure 1 is misleading. The graphs are described as (A) PDIA3 knockout and (B) 1,25(OH)2D3. This suggests that (B) shows wild type cells.

4.       The results presented in Figures 2 A and 2B show opposite effects for PDIA3 knock-out cell line exposed to 1,25(OH)2D3 when measured in two different methods. It looks like the problem with methodology, as both methods measure intracellular calcium levels.

5.       In Figure 2E the TRPV band are cut in such way that they are not visible.

6.       The same problem occurs in some blots from figure 4.

7.       The total levels of PLAA, PLCγ and PKCα are not correct to show activation of these proteins. Other methods, such as recruitment of PLCγ to the cell membrane should be used.

Minor problems:

1.       Full names of proteins or compounds followed by the abbreviations should be presented at the first description, also in the abstract.

2.       Line 45: what does steroid protein mean?

3.       Line 59: PKC is a Protein Kinase C not the phosphokinase C.

4.       Line 74: Dot is unnecessary in 1,25(. OH)2D3.

5.       Line 189: In “calcium influx calcium” one calcium is too much.

6.       Lines 275-276: Something is missing in this sentence.

7.       Line 305: In numerous s

8.       Study should be corrected to studies.

9.       Formatting of some references is wrong, for example 921) and (22).

Comments on the Quality of English Language

In addition to above mentioned errors, some "a"s and "the"s are missing.

Author Response

Reviewer 1

Dear Reviewer,

We appreciate all your comments. We believe that your suggestions significantly improved our manuscript

The paper by Nowak et al. describes the results of VDR or PDIA3 gene knock-outs in human basal cell carcinoma cell line (A431). The authors show that in comparison to wt-A431 cells, A431ΔPDIA3 cells respond differently to 1,25(OH)2D3 (the active form of vitamin D3). In this paper the authors concentrate on calcium transport and signalling.

Despite there are some interesting results in this paper, there are several issues that must be corrected before the paper is suitable for publication.

Major problems:

  1. 1,25(OH)2D3 was used only at one, very high concentration of 100nM. Why so high concentration? Were lower concentrations tested as well?

Thank you for the question. The following explanation was added into the materials and methods section 1,25(OH)2D3: “100 nM concentration was used in all experiments, as it was proven to have most potent activity for proliferation as well as for photoprotection in other research. In previous publications 3, 4, we tested different concentrations using proliferation assay (10-6-10-12M) and 100nM concentration was selected due to optimal activation of 1,25(OH)2D3-regulated pathways in our cell model. Additionally, in the case of the calcium influx probe (Fura-2AM), we tested other concentrations, 1µM, and 10nM, but we didn’t observe significant changes in our results regarding the A431∆PDIA3 cell line.” Supplementary Figure 1 with data has been added to the manuscript. Consequently, the reference to Supplementary Figure 1 was added.

  1. PDIA3 knock-out cell line was described in the Materials and methods, but VDR knock-out cell line was not.

Thank you for the comment. Appropriate information was included in the final manuscript.

  1. The description of Figure 1 is misleading. The graphs are described as (A) PDIA3 knockout and (B) 1,25(OH)2D3. This suggests that (B) shows wild-type cells.

Thank you very much for the comment. The description of figure 1 was clarified and changed in the manuscript to: “change in expression level after (A) PDIA3 knockout and (B) after 1,25(OH)2D3 treatment solely in A431∆PDIA3 cells”.

  1. The results presented in Figures 2 A and 2B show opposite effects for PDIA3 knock-out cell line exposed to 1,25(OH)2D3 when measured in two different methods. It looks like the problem with methodology, as both methods measure intracellular calcium levels.

Thank you for the remark. We performed both experiments as we wanted to make sure that the observed effect would be comparable. Both probes were measured with the use of different methods; Fura-2AM on plate reader and Fluo-4AM with fluorescence microscopy. The difference might be caused by the material cells were seeded into, as Fura-2AM was performed at 96-well polystyrene plates and Fluo-4AM on glass chamber slides. We observed that A431∆PDIA3 cells display a different morphology than other A431 cell lines. Further, the lowered cell-surface adhesion was observed within this subline, especially on glass chamber slides. That might be caused by the knockout of PDIA3, as it changed the expression of adhesion molecules 3. However, data from both probes demonstrate that 1,25(OH)2D3-induced calcium influx after PDIA3 knockout is impaired.

  1. In Figure 2E the TRPV band are cut in such way that they are not visible.

Thank you for your comment. Photographs of bands have been adjusted within Figure 2 for better quality and visibility. Corrected Figure 2 was attached to the manuscript.

  1. The same problem occurs in some blots from figure 4.

Thank you for your comment. Photographs of bands have been adjusted within Figure 4C for better quality and visibility. Corrected figure 4 was attached to the manuscript.

  1. The total levels of PLAA, PLCγ and PKCα are not correct to show activation of these proteins. Other methods, such as recruitment of PLCγ to the cell membrane should be used.

Thank you for your valuable suggestion. Indeed, it would be beneficial to see activation of those proteins, however for the time being we do not have funding to perform additional experiments.

“On the other hand, deletion of PDIA3 had the most prominent effect on the expression of PLCγ, which might partially explain impairment in calcium mobilization in A431∆PDIA3 cells, as activation of PLCγ-mediated signaling results in the release of calcium from intracellular stores and induce influx of this ions across plasma membrane 7. Decreased response levels of PKCα to 1,25(OH)2D3 in A431∆PDIA3 cells lie in agreement with the observation of Boyan and colleagues where partial deletion of PDIA3 impaired activation of PKC through PLAA induction due to lack of interaction between PDIA3/Cav1/PLAA 8”.

Minor problems:

  1. Full names of proteins or compounds followed by the abbreviations should be presented at the first description, also in the abstract.

Thank you for the comment. All abbreviations were developed and added to the manuscript.

  1. Line 45: what does steroid protein mean?

Thank you for the comment.  I’m sorry but that was a mistake in abbreviation development. It was corrected within the manuscript.

  1. Line 59: PKC is a Protein Kinase C not the phosphokinase C.

  1. Line 74: Dot is unnecessary in 1,25(. OH)2D3.

  1. Line 189: In “calcium influx calcium” one calcium is too much.

Thank you for the comments. Issues 3-5 were corrected in the manuscript.

  1. Lines 275-276: Something is missing in this sentence.

Thank you for the remark. The sentence was reviewed and rewritten for clarity.

  1. Line 305: In numerous s Study should be corrected to studies.

The correction has been made in the manuscript.

  1. Formatting of some references is wrong, for example 921) and (22).

Thank you for the remark. The format of references was fixed and references were once more verified for any errors.

Comments on the Quality of English Language

In addition to above mentioned errors, some "a"s and "the"s are missing.

As for the comment related to the English language, the manuscript was once more verified for linguistic errors and we hope it is acceptable in its present form.

(1) Chaiprasongsuk, A.; Janjetovic, Z.; Kim, T. K.; Jarrett, S. G.; D'Orazio, J. A.; Holick, M. F.; Tang, E. K. Y.; Tuckey, R. C.; Panich, U.; Li, W.; et al. Protective effects of novel derivatives of vitamin D(3) and lumisterol against UVB-induced damage in human keratinocytes involve activation of Nrf2 and p53 defense mechanisms. Redox Biol 2019, 24, 101206. DOI: 10.1016/j.redox.2019.101206  From NLM.

(2) Tuckey, R. C.; Li, W.; Shehabi, H. Z.; Janjetovic, Z.; Nguyen, M. N.; Kim, T. K.; Chen, J.; Howell, D. E.; Benson, H. A.; Sweatman, T.; et al. Production of 22-hydroxy metabolites of vitamin d3 by cytochrome p450scc (CYP11A1) and analysis of their biological activities on skin cells. Drug Metab Dispos 2011, 39 (9), 1577-1588. DOI: 10.1124/dmd.111.040071  From NLM.

(3) Nowak, J. I.; Olszewska, A. M.; Piotrowska, A.; MyszczyÅ„ski, K.; Domżalski, P.; Å»mijewski, M. A. PDIA3 modulates genomic response to 1,25-dihydroxyvitamin D(3) in squamous cell carcinoma of the skin. Steroids 2023, 199, 109288. DOI: 10.1016/j.steroids.2023.109288  From NLM.

(4) Piotrowska, A.; Wierzbicka, J.; Åšlebioda, T.; Woźniak, M.; Tuckey, R. C.; Slominski, A. T.; Å»mijewski, M. A. Vitamin D derivatives enhance cytotoxic effects of H2O2 or cisplatin on human keratinocytes. Steroids 2016, 110, 49-61. DOI: 10.1016/j.steroids.2016.04.002  From NLM.

(5) PÅ‚udowski, P.; Kos-KudÅ‚a, B.; Walczak, M.; Fal, A.; ZozuliÅ„ska-ZióÅ‚kiewicz, D.; Sieroszewski, P.; Peregud-Pogorzelski, J.; Lauterbach, R.; Targowski, T.; LewiÅ„ski, A.; et al. Guidelines for Preventing and Treating Vitamin D Deficiency: A 2023 Update in Poland. Nutrients 2023, 15 (3). DOI: 10.3390/nu15030695  From NLM.

(6) Souberbielle, J. C.; Cavalier, E.; Delanaye, P.; Massart, C.; Brailly-Tabard, S.; Cormier, C.; Borderie, D.; Benachi, A.; Chanson, P. Serum calcitriol concentrations measured with a new direct automated assay in a large population of adult healthy subjects and in various clinical situations. Clin Chim Acta 2015, 451 (Pt B), 149-153. DOI: 10.1016/j.cca.2015.09.021  From NLM.

(7) Gusev, K.; Glouchankova, L.; Zubov, A.; Kaznacheyeva, E.; Wang, Z.; Bezprozvanny, I.; Mozhayeva, G. N. The store-operated calcium entry pathways in human carcinoma A431 cells: functional properties and activation mechanisms. J Gen Physiol 2003, 122 (1), 81-94. DOI: 10.1085/jgp.200308815  From NLM.

(8) Boyan, B. D.; Chen, J.; Schwartz, Z. Mechanism of Pdia3-dependent 1α,25-dihydroxy vitamin D3 signaling in musculoskeletal cells. Steroids 2012, 77 (10), 892-896. DOI: 10.1016/j.steroids.2012.04.018  From NLM.

Reviewer 2 Report

Comments and Suggestions for Authors

The manuscript „VDR and PDIA3 are essential for activation of calcium signaling and membrane response to 1,25(OH)2D3 in squamous cell carcinoma cells” presents the effects of knockout of PDIA3 and VDR on 1,25(OH)2D3 membrane, non-genomic signaling.

1. The manuscript is prepared correctly, contains clear methods description.

2. The manuscript contains a lot of information and is interesting.

3. Figures are well described in the text and well signed. Their quality is good.

4. The discussion contains a lot of information in relation to the obtained research.

Question:

Have the authors determined a toxic dose of 1,25(OH)2D3 for the tested cell line?

Suggestions/comments:

1.      The abstract should have a different form, specifying background, methods, results etc.

2.      Line 74 – correct subtitle

3.      Line 383 – is the reference correct?

Author Response

Reviewer 2

Dear Reviewer,

Thank you very much for your comments, which allow us to improve our manuscript.

The manuscript „VDR and PDIA3 are essential for activation of calcium signaling and membrane response to 1,25(OH)2D3 in squamous cell carcinoma cells” presents the effects of knockout of PDIA3 and VDR on 1,25(OH)2D3 membrane, non-genomic signaling.

  1. The manuscript is prepared correctly, contains clear methods description.

  1. The manuscript contains a lot of information and is interesting.

  1. Figures are well described in the text and well signed. Their quality is good.

  1. The discussion contains a lot of information in relation to the obtained research.

Question:

Have the authors determined a toxic dose of 1,25(OH)2D3 for the tested cell line?

Thank you for the question. The toxicity of 1,25(OH)2D3 was established in our previous work 3 with SRB assay in A431 cell lines. 1,25(OH)2D3 at 1µM concentration reduced viability of cell about 20%, while 100nM reduced cell viability around 15% in A431 wild-type cells, while VDR and PDIA3 knock-out cell line were more sensitive to lower concentrations of 1,25(OH)2D3.

Suggestions/comments:

  1. The abstract should have a different form, specifying background, methods, results etc.

Thank you for your suggestion. The proposed form of abstract has been applied.

  1. Line 74 – correct subtitle

  1. Line 383 – is the reference correct?

Thank you for your remark. The references have been once more reviewed for errors and corrected within the manuscript.

Round 2

Reviewer 1 Report

Comments and Suggestions for Authors

In the revised version of the paper, the authors introduced some of the requested changes. Unfortunately, some of the problems remain, and need to be corrected.

Major issues:

In Figure 2E the TRPV bands are exactly as they were in the previous version.

Since the methods to measure activation of PLAA, PLCγ and PKCα are wrong, and the Authors can’t use the correct methods, the whole section must be removed. In such situation the text and the Figure 5 must be adjusted to the data obtained by the Authors in a correct way.

Author Response

Major issues:

  1. In Figure 2E the TRPV bands are exactly as they were in the previous version.

Thank you for that remark, the picture of TRPV6 bands on figure 2E was replaced with an improved according to reviewers suggestions. We would like to apologize for this omission.

  1. Since the methods to measure activation of PLAA, PLCγ and PKCα are wrong, and the Authors can’t use the correct methods, the whole section must be removed. In such situation the text and the Figure 5 must be adjusted to the data obtained by the Authors in a correct way.

That's an excellent point, thank you. Fortunately, we planned to measure PLA2 and PKCα activity and were prepared for it. Cell lysates were freshly prepared and we use ELISA to measure activity. PLA2 activity in A431 was very weak, while PKCα activity was very strong. Unfortunately, no significant changes were observed after 1,25(OH)2D3 treatment of wild-type A431 cells or A431 cells with VDR or PDIA3 deletion. Therefore, we decided to modify the manuscript in accordance with the reviewer's suggestion. Figures 4 and 5 have been modified (also graphical summary). Western blot showing PLAA, PLCγ and PKCα levels were transferred to the supplement. The abstract, results, and the discussion have been revised accordingly.